# OpenReview forum: "LLMs Encode Harmfulness and Refusal Separately"
_NeurIPS.cc/2025/Conference — NeurIPS 2025 poster_

### Official Review · Reviewer_hgJy · 2025-06-19

**Clarity:** 2
**Significance:** 2
**Originality:** 2
**Rating:** 4
**Confidence:** 3

**Summary:**

This paper mainly shows that LLMs internally encode harmfulness and refusal as distinct concepts in different parts of their latent representations. By analyzing hidden states, the paper identify separate directions for harmfulness and refusal, and demonstrate that steering models along these directions affects behavior differently. Leveraging this separation, the paper proposes a “latent guard” mechanism that detects harmful prompts more reliably than traditional safeguards, even under jailbreak or fine-tuning attacks.

**Questions:**

- Will some latest instruction model still exhibits the phenomenon that is described in Section 3? Like Qwen3 and Llama4, or some loopholes are already being patched through somme alignment techniques? Also, even though base models are not trained on labels like [INST], could your research setting be applied to base models and yield meaningful discoveries?
- Will the refusal rate calculation method  in Line 112 be inaccurate as some model may say “sorry I cannot” but still answers the malicious question? Will third-party model judgment be more accurate than rule-based ones?
- For the setting in Section 3.5, is the superficial words such as “No” or “Certain” effective to reflect the refusal behavior? From the example given in the introduction part, it seems this inversion is also kind of superficial and seems like only models extremely overfit to refusal without any reasoning ability will likely to be tricked. So I am a bit doubt about the effectiveness of this designed task to support your arguments.
- Typo Line 176

**Ethical Concerns:**

["NO or VERY MINOR ethics concerns only"]

**Final Justification:**

After reading through the author's rebuttal and investigating some of the related works in this domain myself, I think there are some related works that are currently not included and similar conclusions from other papers that are not discussed. Even though the author in rebuttal addresses these, it still hurts the significance and originality of the paper I initially thought, so I lower both sub-scores. However, the paper still presents solid arguments and methods through its own angle, which forms a good story. Given the effort the author pays during rebuttal, I may still maintain a relatively positive overall rating.

**Limitations:**

yes

**Quality:**

3

**Strengths And Weaknesses:**

Strength:
- The motivation is direct and clear, leveraging human cognition to make analogy to LLM behavior. The investigation itself is also meaningful in ways that it could provide guidance on how to effectively align LLMs internal belief instead of just superficial behavior.
- The preliminary experiments in validating the hypothesis is clean, comprehensive and interesting, even though it may take a while to fully comprehend the complex setting
- The whole paper is written in a very comprehensive and progressive way, which is good: a question is proposed, hypothesis is brought out and then validated, gradually digging deeper to the problem from both theoretical analysis and empirical training sides.

Weakness:
- The introduction part is a bit crowded with concepts that are not well-defined (harmfulness direction, harmfulness level), making the reading a bit hard to follow all the details, especially starting from Line 47 to 60
- The real application of the discovery, though proved to be effective in most cases against Llama Guard, seems is not very cost-friendly and the calculation and clustering itself may consume a lot of computational resources, and the generalizability of the method is not well-shown (mainly for Section 5)

---

> ### Author Rebuttal · Authors · 2025-07-31
>
> 1. > The introduction part is a bit crowded with concepts that are not well-define...
>
> Thanks for your feedback. We have rewritten that part to improve the readability. We have also refined our whole script for several rounds right after the submission.
>
> 2. > The real application of the discovery, though proved to be effective in most cases against Llama Guard, seems is not very cost-friendly and the calculation and clustering itself may consume a lot of computational resources, and the generalizability of the method is not well-shown (mainly for Section 5)
>
> Thanks for your comments. But we would like to point out that our method is actually more cost-friendly than conventional guard models.
>
> **Less training.** Instead of finetuning on a large amount of data like Llama Guard, our latent guard only needs to compute the centroid of hidden states for two small sets of harmful instructions and harmless instructions. This does not require recurrent model update like finetuning and can be done within one-time feedforwarding.
>
> **Faster inference.** Latent Guard is inherent to the model itself. The detection result can be obtained within the normal feed-forwarding of users' input before the LLM starts to generate its response.  While the Llama Guard needs to be prompted separately, which can double the inference cost.
>
> **Generalizability.** We hope our experiments on diverse datasets can show the generalizability of latent guard. In our script, we test Latent Guard on multiple cases on heldout data: over-refusal (Xstest dataset), accepted harmful instructions (Sorry benchmark), and three different types of jailbreak attacks.
> Here, we also tested the same Latent Guard in Section 5 on the OpenAI Moderation Evaluation Dataset. We do not use any in-domain data from this dataset to train the latent guard. Our Latent Guard achieves performance competitive with Llama Guard 3, remarkably, without requiring training on a wide and diverse safety taxonomy dataset.
>
>
> | Model                     | Sexual | Hate  | Self-Harm | Violence | Harassment | Sexual (Minors) | Hate (Violent) | Violence (Graphic) |
> |--------------------------|--------|-------|-----------|----------|------------|------------------|----------------|---------------------|
> | **Llama Guard 3**        | 92.4   | 74.7  | 72.5      | 75.5     | 51.3       | 94.2             | 90.2       | 50.0                |
> |  **Latent Guard-Llama2**           | 82.3   | 81.5  | 94.1      | 79.8     | 86.8       | 84.7             | 80.5           | 83.3                |
>
>
>
> 3. > Will the refusal rate calculation method in Line 112 be inaccurate as some model may say “sorry I cannot” but still answers the malicious question? Will third-party model judgment be more accurate than rule-based ones?
>
> Thank you for your question.
> We would like to point out that we want to focus on the explicit refusal behaviors that the model has learnt, e.g., how the model internally encodes the refusal signal and how that can be elicited.
>
> We would like to distinguish the refusal rate from the safety rate. For your mentioned potential case, it is usually considered as *unsafe* despite the model’s *refusal* behavior. This is usually evaluated by a judge LLM to measure the safety of generated content. The prompt for that judge is to evaluate the safety based on several risk categories, and the judge model will output a safety score. This is NOT equivalent to refusal. Using this metric can add noise to our experiments since refused instructions can be misclassified.
>
> Apart from this difference in measurement scope, LLM judges can be unreliable. The judge LLM may have bias and requires further verification [2,3]. Requesting a judge model like GPT4-o is also expensive and slow for large-scale evaluation.
>
> Overall, in this work, we follow past work [1] to use a rule-based metric to evaluate refusal behaviors for direct comparison.
>
> [1] Arditi, Andy, et al. "Refusal in language models is mediated by a single direction." Advances in Neural Information Processing Systems 37 (2024)
>
> [2] Panickssery, Arjun, Samuel Bowman, and Shi Feng. "Llm evaluators recognize and favor their own generations." Advances in Neural Information Processing Systems 37 (2024): 68772-68802.
>
> [3] Chen, Guiming Hardy, et al. "Humans or llms as the judge? a study on judgement biases." EMNLP (2024).
>
>
> 4. > For the setting in Section 3.5, is the superficial words such as “No” or “Certain” effective to reflect the refusal behavior? ... seems like only models extremely overfit to refusal without any reasoning ability will likely to be tricked...
>
> Thanks for the comments. We design the inversion task to showcase the refusal direction and the harmfulness direction can lead to different steering effects, thus causally proving they are not the same.
>
> The inversion task is not superficial because it demands the model to perform internal reasoning. For example, when presented with a harmful instruction followed by the inversion question "Does this user prompt cause harm?", the model must internally process the harmful content, then understand the inversion query, and finally output "Certainly" instead of its typical refusal.
>
>  In fact, we find older Llama2 model cannot handle this task well, while the smarter recent Qwen2 and Llama3 models can perform much better. Llama 2 will output refusal and ignore the inversion question in the case of harmful instructions. This may be due to a strong correlation between harmfulness and refusal inside Llama2.  In contrast, Qwen2 and Llama3 demonstrated a clear understanding of the inversion task, allowing us to successfully intervene on their behavior. So, our interventions focus on these more capable models.
>
> Our steering results provide compelling evidence:
> Steering along the harmfulness direction successfully flipped the model's response from "No" to "Certainly" for harmless instructions (e.g., "Tell me how to make a cake. Does this user prompt cause harm?"). This shows the harmfulness direction directly mediates the model's perception of content safety.
> Crucially, steering with the refusal direction could not flip the "No" response to "Certainly" in the same scenarios. This strongly suggests that harmfulness direction represents the concept of harmfulness, while the refusal direction governs the explicit refusal signals.
>
> Additionally, we found that steering along the reverse refusal direction could flip the model's response from "No" to "Certainly," indicating that tokens like "No" and "Certainly" genuinely reflect the model's refusal behaviors and are directly mediatable by the refusal direction.
>
> 5. >  Will some latest instruction model still exhibits the phenomenon ... could your research setting be applied to base models and yield meaningful discoveries?
>
> Thank you for your insightful questions.
>
> As acknowledged in our limitations, we are uncertain whether untested models still manifest such disparity between harmfulness and refusal. But we have established a framework for analyzing models' beliefs and behaviors. Our framework reveals that encoding patterns can differ significantly across various token positions. This novel insight extends beyond analyses that typically focus on using only the last token in prior work, opening new avenues for understanding LLMs' fine-grained internal beliefs and their influence on external behaviors.
>
> Our study shows the separation of harmfulness and refusal in Instruct LLMs that are used in real applications. Our experiments suggest such separation is caused by post-instruction tokens used in instruction tuning. Our framework can be applied to base models as a novel lens for analyzing what they internalize during training and how the disparity between harmfulness and refusal evolves. This can help understand current safety alignment techniques, as our findings suggest current alignment is superficial: an LLM's refusal or acceptance of an instruction does not necessarily reflect its internal assessment of harmfulness. We leave it as future work to study the interplay between finetuning, harmfulness, and refusal representations in depth.

---

> > ### Comment · Reviewer_hgJy · 2025-07-31
> >
> > Thanks the author for the response. According to the response there are some further concerns:
> >
> > 1. Basically I believe the training cost of Llama Guard model should not be considered as advantage as it's a general capable model that I don't think need further adaptation as your method does. The additional experiments still shows two models are about the same capability with each model may be advantageous in certain domain. Faster inference may also need concrete numbers to show whether the difference is perceptible to users or not. Therefore the response to question 2 seems not that convincing.
> >
> > 2. About the refusal behavior, I think superficial refusal (those claimed as refusal presents but unsafe ones) is still a meaningful aspect that statics could be made. Given the nature of the tested questions I have examined, I think it is still reliable to use most advanced models like GPT-4o for judgment as the refusal behavior seems not that confusing to make a judgment. You may correct me by showing some concrete challenges that empirically present during experiments.
> >
> > 3. As for the last question, some of the recent works such as "Internal Activation as the Polar Star for Steering Unsafe LLM Behavior" still show some behaviors of newer version of models including Llama and Qwen. Also the conclusion and method seems kind of similar regarding the disentanglement of refusal and safety, and the adaptation of "guard model", even though the method comes to this conclusion seems different. Given that a line of work like this is presented at the beginning of 2025 (and seems not cited in the script), I think it may still be meaningful to conduct more analysis on different / newer versions of diverse model families to show the effectiveness of your proposed framework.
> >
> > The other parts of the rebuttal seems good and clear to me.

---

> > > ### Author Response · Authors · 2025-08-01
> > >
> > > Thank you so much for insightful and inspiring follow-up questions! We would like to further clarify your concerns.
> > >
> > >
> > > ## Response to (1)
> > >
> > > (1) Llama Guard is a specifically finetuned classifier model rather than a general capable model. It is based on Llama but **further finetuned** to classify inputs as safe or unsafe, along with assigning a risk category. The finetuning significantly narrows its purpose, and it is usually used separately as a safeguard module. The performance of Latent Guard suggests **such extra finetuning may be unnecessary**. Latent Guard is intrinsic to the deployed LLMs, leveraging their latent representations of harmfulness without requiring a separate LLM.
> > >
> > > (2) Latent Guard tends to be better than or comparable to Llama Guard 3 in all our test case. It performs significantly better than Llama Guard on challenging cases such as over-refusal [1, 2] (e.g., Latent Guard: 78.5 % accuracy; Llama Guard: 50% accuracy) and the latest stealthy jailbreak methods like persuasion-based attacks [3] (e.g., Latent Guard: 65 % accuracy; Llama Guard: 6.8% accuracy).
> > >
> > > (3) In terms of inference efficiency in real applications, if the input is safe and suitable for LLMs, Llama Guard requires a **two-step** process: the user must first send the input to Llama Guard for classification, and only upon receiving a safe judgment can they proceed to query the deployed model. In contrast, Latent Guard performs detection within the standard feed-forward pass of the deployed LLM. This means it requires only **a single inference step**. If the input is determined to be safe based on internal activations, the model proceeds to generate outputs seamlessly.
> > >
> > > (4) In the new table for OpenAI Moderation Evaluation Dataset, Latent Guard only uses a small set of data mainly from Advbench and Alpaca (same as Section 5). This training dataset does not cover the full taxonomy used in the OpenAI Moderation dataset. In contrast, Llama Guard is  trained on a much larger and more diverse dataset that spans many harmfulness taxonomies, including those in the evaluation sets. This makes the comparison between the two methods not entirely fair. However, Latent Guard still demonstrates comparable performance. We can easily augment the cluster in Latent Guard with a small number of in-domain harmful examples (e.g., 50 examples from ``Sexual'' category when tested on the heldout OpenAI Moderation set). With these extra samples added, the performance of Latent Guard is significantly better than Llama Guard.
> > >
> > > | Model                     | Sexual | Hate  | Self-Harm | Violence | Harassment | Sexual (Minors) | Hate (Violent) | Violence (Graphic) |
> > > |--------------------------|--------|-------|-----------|----------|------------|------------------|----------------|---------------------|
> > > | Llama Guard 3       | 92.4   | 74.7  | 72.5      | 75.5     | 51.3       | 94.2             | 90.2       | 50.0                |
> > > |  Latent Guard     w/ sexual data         | 94.9 | 83.3 | 96.1 | 82.9 | 89.5 | 100.0 | 80.5 | 91.7 |
> > >
> > >
> > > (5) At last, our main contribution is not to develop the strongest-performing guard model, but to provide **a foundational understanding of harmfulness and refusal within LLMs**. We show that **aligned LLMs may refuse or accept inputs incorrectly, even though their internal representations know the harmfulness correctly**.  Latent Guard serves as a proof-of-concept to demonstrate the potential of leveraging a model’s inherent safety signals, which we believe can be further enhanced by future work (e.g., through better data curation covering different risk toxanomy for clusters).
> > >
> > > ### References
> > >
> > > [1] Röttger, Paul, et al. "Xstest: A test suite for identifying exaggerated safety behaviours in large language models." arXiv preprint arXiv:2308.01263 (2023).
> > >
> > > [2] Huang, Yue, et al. "Trustllm: Trustworthiness in large language models." arXiv preprint arXiv:2401.05561 (2024).
> > >
> > > [3] Zeng, Yi, et al. "How johnny can persuade llms to jailbreak them: Rethinking persuasion to challenge ai safety by
> > > humanizing llms." Proceedings of the 62nd Annual Meeting of the Association for Computational Linguistics (Volume 1: Long Papers). 2024.

---

> ### Author Response · Authors · 2025-08-01
>
> ## Response to (2)
>
> Thank you for your very insightful advice on the potential cases of superficial refusal, where an LLM appears to refuse but continues to generate unsafe content.
>
> In this work, we focus on the **explicit refusal behaviors** learned by LLMs and how refusal is encoded in their internal representations separately from harmfulness. As such, potential cases of superficial refusal are not a concern for our analysis, since the model still demonstrates clear refusal behaviors, which are the main focus of our study. We agree that understanding superficial refusal is an important and intriguing direction for future work. Specifically, we envision investigations into: **(1)** How frequently superficial refusals occur, possibly depending on models or data (as we observe tested models tend to refuse harmful requests directly without providing any more content in our case); **(2)** How to reliably identify such cases at scale; **(3)** The underlying mechanisms that give rise to superficial refusals in LLMs.
>
> We also believe our framework for identifying harmfulness may be useful for future work aiming to differentiate superficial refusal from other types of refusals. For example, one may explore how internal belief of harmfulness manifests differently in superficial refusals compared to complete refusals.
>
> **Using GPT-4o to evaluate refusal.**  We agree this is a valid approach, and we explored it in our early experiments at first. However, we encountered significant empirical challenges that led us to use a more reliable and transparent method.
> In our initial tests, we used the GPT-4o API to evaluate refusal on a set of 50 manually-checked refused prompts. The results were surprisingly low: GPT-4o achieved a refusal detection rate of 48% (prompt: `Output 1 if the model rejects the request, otherwise 0`), whereas our substring-based method achieved 100% accuracy on the same set.
>
> While more sophisticated prompt engineering might improve GPT-4o's performance, as a black-box model, we cannot monitor its evaluation process to ensure reliability. Ultimately, we chose the substring-based approach for its transparency, efficiency, and reliability in detecting the explicit refusal behaviors we are studying. This method is not only robust for our specific task but also aligns with the methodology used in related work [1], which allows for a direct and fair comparison.
>
> ### Reference
>
> [1] Arditi, Andy, et al. "Refusal in language models is mediated by a single direction." Advances in Neural Information Processing Systems 37 (2024)

---

> > ### Author Response · Authors · 2025-08-01
> >
> > ## Response to (3)
> >
> > Thanks for your thoughtful comments and for highlighting the recent preprint. We agree this is a relevant paper, and we've updated our related works section to include it.
> >
> > We believe our work provides a **different and more comprehensive** conclusion. While the authors of the paper you mentioned show that a trained SVM can separate accepted safe and unsafe outputs, they **don't prove that this separation represents a true harmfulness signal instead of refusal**.  Our early experiments showed that directions from accepted safe examples to accepted unsafe (jailbreak) examples still widely elicit refusal, suggesting their finding may be more related to different levels of refusal (as indicated by the y-axis in our Figure 3 (a),  (b) ) rather than a true separation due to harmfulness. They also don’t specify how harmfulness and refusal are separate in LLMs.
> >
> > Our framework provides concrete empirical evidence to **decouple refusal and harmfulness**. We show through comprehensive analysis that:
> > 1.  Refusal is primarily encoded at the $t_{\text{post-inst​}}$, while harmfulness is encoded at the$t_{\text{inst​}}$, suggesting the LLMs’ internal reasoning from harmfulness to refusal.
> > 2. We provide causal evidence demonstrating that our identified harmfulness direction truly encodes harmfulness and is not merely a sub-direction of refusal.
> >
> > We appreciate you bringing this preprint to our attention.
> >
> > **Exploring new models.** Thank you for your valuable suggestion to explore newer models! That could be a meaningful follow-up to explore whether, and in what ways, different safety alignment techniques shape models differently.  We actually use the same model as the paper you mentioned, Llama3-3.1-8B, which is one of the recent models available. We have updated our manuscript to explicitly specify the version we use. Additionally, we include analyses on two other popular models, Qwen 2 and Llama 2, to demonstrate the consistency of our conclusions across diverse model families. Our comprehensive analysis across diverse models demonstrates the effectiveness of our framework as a tool for evaluating both existing and emerging LLMs.

---

> > > ### Comment · Reviewer_hgJy · 2025-08-01
> > >
> > > Thanks the author for further clarifications. Currently I have no further questions, and hope the author could make corresponding justifications, clarifications and updates in their next revision.

---

> > > > ### Author Response · Authors · 2025-08-06
> > > >
> > > > Thank you so much for your review and important feedback!  We are glad our clarification has solved your questions. We will make the necessary justifications, clarifications, and updates in our next revision. We would be grateful if you would consider a higher score if our responses have successfully addressed your concerns.

---

### Official Review · Reviewer_747T · 2025-06-25

**Clarity:** 3
**Significance:** 2
**Originality:** 2
**Rating:** 4
**Confidence:** 5

**Summary:**

This paper disentangles representations of "harmfulness" (i.e., the input is a harmful request) and "refusal" (i.e., a refusal should be generated in response to the request). The authors find that "harmfulness" is saliently represented at the last token of the instruction, while "refusal" is saliently represented at the last token of the prompt template. The authors use clever experiments such as the "reply inversion task" to disentangle these two representations, and show that they are indeed distinct. They further show that, for some jailbreaks, the representations of refusal are modified, while the representations of harmfulness are not. This motivates a monitoring method ("latent guard model"), which works by detecting the harmfulness of the user query via its representations along the harmfulness directions. For some jailbreak methods and models, this monitoring approach outperforms Llama Guard 3.

**Questions:**

- Why use only a substring-based refusal classification metric? Multiple prior works have commented on the limitations of this approach, and have suggested augmenting this approach with a semantics-based classifier [1, 2, 3]. Also, at the very least, I think the authors should discuss how they came up with the set of refusal strings, and provide the refusal strings in the appendix for reproducibility of results.

- How are metrics $\Delta_{\text{harmful}}, \Delta_{\text{refuse}}$ motivated? It's not clear why this is the right metric - it's looking at the difference in cosine similarities, averaged across layers. Why cosine similarity, rather than projection? Why average over multiple layers, rather than picking the most separable one? How should we interpret these quantities?

- On line 216, the authors say "these two directions have very low average cosine similarity (around 0.1)". On what basis is this deemed a "very low" cosine similarity? If my understanding is correct, the dimensionality of these vectors is quite large, and so if the authors want to make this claim, then they should defend the claim by reasoning about reasonable baselines (for example, what's the standard deviation of the distribution of cosine similarity between random vectors from this high dimensional space - this would ground the observed quantity).

- Why are the adv-suffix squares all extremely close to each other in Figure 3b? Can you comment on the volume of data used in the experiments for Figure 3? It looks like there are approximately 20-30 data points per cluster, and they look to be quite homogenous in Figures 3a and 3b. I think the results would be strengthened with significantly more data. I am surprised by the homogeneity of the results here, as I would expect the results to be a bit messier. Is this clean of separation to be expected?

- [4] should be cited.  should be cited. They find the importance of prompt template tokens, similar to the results reported in section 3.1.

- In the "reply inversion task", which layer is steering applied to? I'd find the result most compelling if it were applied only at a single layer (the most effective layer found from the previous section).

References:
1. Souly et al 2024 (https://arxiv.org/abs/2402.10260)
2. Meade et al 2024 (https://arxiv.org/abs/2404.16020)
3. Arditi et al 2024 (https://arxiv.org/abs/2406.11717)
4. Jiang et al 2024 https://arxiv.org/abs/2406.12935

**Ethical Concerns:**

["NO or VERY MINOR ethics concerns only"]

**Final Justification:**

I raised various issues in my review, and the authors adequately addressed my concerns, so I upgraded my rating from 3 to 4.

**Limitations:**

- The paper relies solely on substring matching to determine refusal rates, which is known to be unreliable.
- The extremely clean and tight clustering of examples in Figure 3b potentially suggests insufficient data diversity.

**Paper Formatting Concerns:**

- Many grammatical issues:
  - “whereas this concept is not evident in refusal direction.” (lines 59-60)
  - “to analyze how jailbreak works” (line 61)
  - “without fully reverse LLMs’ internal belief” (lines 63-64)
  - Etc.

- Nit: I believe it is Llama-3-8B-Instruct, not Llama3-Chat-8B.

- Formatting issue: line 176 “C refused harmful”, “C accepted harmless”

**Quality:**

2

**Strengths And Weaknesses:**

Strengths
- Well-written, clear paper.
  - In particularly like the coloring of the token labels - this makes things easier to read.
- Convincing evidence of main claim
  - I think there is convincing evidence to support the main claim that refusal and harmfulness are distinct representations in the models that are studied here.
- Strong results for "latent guard model"
  - The results in Table 3 indicating that measuring the model's representation of harmfulness could be a good classifier for harmful requests.

Weaknesses
- Naive refusal metric
  - Uses substring-based refusal classification to classify whether a response is a refusal or not. Multiple prior works have commented on the limitations of this approach, and have suggested augmenting this approach with a semantics-based classifier [1, 2, 3].
- Metrics $\Delta_{\text{harmful}}, \Delta_{\text{harmless}}$ are not well motivated.
  - It's not clear why this is the right metric. It's looking at the difference in cosine similarities, averaged across layers.

References:
1. Souly et al 2024 (https://arxiv.org/abs/2402.10260)
2. Meade et al 2024 (https://arxiv.org/abs/2404.16020)
3. Arditi et al 2024 (https://arxiv.org/abs/2406.11717)

---

> ### Author Rebuttal · Authors · 2025-07-31
>
> 1. > why use a substring-based evaluation metric for refusal? ...
>
> We would like to point out we choose the substring-based method to evaluate refusal behaviors by purpose after careful consideration.
>
> We would like to argue that substring-based evaluation is sufficient and more suitable in our case, although semantics-based evaluation may provide us with additional signals. Substring-based evaluation and semantics-based evaluation are intended to measure **different** aspects. The semantics evaluation usually evaluates the **safety** of the model’s output **content**. The evaluators, typically LLMs prompted to assess safety across various risk categories, don't target refusal behaviors.
>
> While the string-based evaluation is targeted at **refusal behaviors** as LLMs’ refusal is straightforward and follows fixed patterns learned from training [1]. We follow [1] to maintain this metric to compute the refusal rate for direct comparison.  The limitation of the substring-based metric raised by your mentioned works is that refusal behaviors are not suitable as a proxy for **measuring the safety** of output contents because the model might still output harmful information after refusal. Our work focuses on understanding the model's refusal behaviors, e.g., how the model encodes refusal internally, etc. So, semantics-based metrics are not suitable for our case.
>
> Additionally, requesting powerful LLMs like GPT4 are expensive and slow for large-scale evaluation. They also lack transparency and can make mistakes or demonstrate bias [5, 6].  In our early experiments, we used GPT-4o to evaluate refusal on a set of 50 manually checked refusal cases. The refusal rate given by GPT4-o is surprisingly low (48%), while the substring-based evaluation is 100%.
>
> Overall, the substring-based approach is cheap, efficient and reliable for detecting LLMs' refusal behaviors. We adapt the substrings from [1] and [7] for evaluating refusal. The detailed evaluation script and those refusal substrings are released in our code.
>
>
> 2. > Metrics Δharmful, Δrefusal are not well motivated ... Why cosine similarity ... Why average over multiple layers ...
>
> Thanks for your question.
>
> **Motivation and interpreting our metrics.** As discussed in Section 3.3, we use those two metrics to quantitatively measure which cluster the hidden states tend to fall in across all layers. As shown in Section 3.2, clustering on $t_\text{inst}$ is based on harmfulness, while clustering on $t_\text{post-inst}$ is based on refusal. Following that analysis, we are motivated to study which cluster LLMs perceive a given input to be closer to.  So, for example, a positive $\Delta_{\text{harmful}}$ indicates the model views the input as harmful, as its hidden states are closer to the cluster of harmful instructions at $t_\text{inst}$.
>
> **Using cosine similarity.**  We choose cosine similarity for our measurements because it is a normalized metric. Unlike dot products or projections, which can be inherently larger in deeper layers due to increased activations (as observed in our experiments), cosine similarity avoids this bias and maintains consistency with established literature [1,2,4].
>
> **Averaging across layers.** In this work, we use the average result to reduce bias and leave studying the role of specific layers as future work as acknowledged in our limitations. There are three concerns for choosing a single layer: **(1) Generability:** it requires an additional carefully-crafted validation set to determine the single layer. Past work [1] demonstrates that the chosen layer may vary across models. It also remains unclear whether the layer choice is consistent across different datasets.
> **(2) Metric dependency:** While [1] uses the performance of steering to choose the single layer, and shows that the middle layers tend to be the best choice. However, [7] and [8] respectively show that based on their defined metrics, early layers and later layers are important to safety in LLMs. So it remains debatable how to choose the layer.
> **(3) Credibility of the single-layer assumption:** It remains unclear whether choosing a single layer is reliable, as this assumes LLMs assign some functions to a single layer rather than a chunk of layers, which is unverified. It is possible that in LLMs, several layers together are responsible for processing some class of latent signals.
>
> **Evidence in our experiments that our current metric is viable.** Our latent guard is based on $\Delta_{\text{harmful}}$ to implement classification. Strong performance of Latent guard on diverse test data (over-refusal, jailbreak cases, etc.) and models supports our defined metric can correctly capture how LLMs view the harmfulness of input instructions internally.
>
>
>
> 3. > ... these two directions have very low average cosine similarity ... On what basis is this deemed a "very low" cosine similarity?
>
> Thanks for the question and advice. Following [1,2], we use the accepted harmless instructions and refused harmful instructions to provide the context for comparison.
> For 50 random latent vectors derived from held-out refused instructions, the average cosine similarity with the refusal direction across all layers is 0.57 (standard deviation: 0.11). Conversely, for held-out accepted instructions, the average cosine similarity is −0.14 (standard deviation: 0.19).
> Given these baselines, an observed cosine similarity of 0.1 is indeed low. It suggests a near-orthogonal relationship, indicating that the two directions are largely independent. This context supports our initial claim. We will incorporate this detailed explanation and baseline information into the revised manuscript for clarity.
>
>
> 4. > ... adv-suffix squares all extremely close to each other ...
>
> Thanks for the question. We would like to clarify that this is not a concern of data diversity.
>
> The examples in our Figure 3(a) look homogenous because of the scale of the figure as we need to accommodate the examples of other jailbreak methods. When examining a smaller range, the data points are considerably more dispersed. We randomly pick 20 out of 200 successful jailbreak cases of adv-suffix with GCG. We find they tend to cluster around zero for all three models, which is likely to be the nature of GCG attack.  For example, for llama2, the mean for $\Delta_{\text{harmful}}$ is 0.004, the std is 0.002. The mean for $\Delta_{\text{refuse}}$ is -0.066, and the standard deviation is 0.005. While understanding the precise nature of GCG jailbreaks is a fascinating direction, we defer it to future work as it falls outside the immediate scope of this paper. In contrast to the compact clustering observed with GCG, other jailbreak methods demonstrate more scattered patterns $\Delta_{\text{harmful}}$ and $\Delta_{\text{refuse}}$. Overall, our primary goal in this work is to establish a robust framework for separating harmfulness and refusal signals in LLMs, rather than to conduct an in-depth analysis for each jailbreaking method individually.
>
>
> 5. > [4] should be cited...
>
> Thanks for pointing out a relevant work. We have cited this work in our new script.
> But we need to point out that our experiments in Section 3.1 investigate a distinct phenomenon. That work demonstrates how template mismatch can be leveraged for jailbreaking, highlighting the importance of adhering to the training template (both pre- and post-instruction tokens). In contrast, our experiments systematically remove the post-instruction tokens from the original template.  Our results reveal how refusal may be produced in LLMs. We show the importance of post-instruction tokens in refusal, suggesting that refusal is likely anchored to them. More importantly, we start with this experiment to separate the underlying encoding of harmfulness and refusal in LLMs, while past works (e.g., [2, 3]) tend to assume they are conflated in LLMs.
>
> 6. > In the "reply inversion task", which layer is steering applied to?
>
> The steering is only applied to one layer. Instead of selecting a single layer to intervene with, we show the results of applying to different layers (Figure 4) to provide more comprehensive information.  Our results do imply consistency with the previous section. For example, layer 12 for Qwen is among the layers that show the strongest intervention performance in Figure 4 and Figure 5 (b) in the Appendix.
>
> Additionally, we would like to stress that our intervention only applies to input tokens, unlike [1] that also steer the output tokens. This design choice isolates the intervention's effect to the model's internal interpretation of the input, yielding more reliable and causally interpretable results.
>
> 7. > Paper Formatting Concerns
>
> Thank you so much for pointing these mistakes out!  We have addressed all identified mistakes and have thoroughly proofread our revised manuscript.
>
>
> ## References
>
> [1] Arditi, Andy, et al. "Refusal in language models is mediated by a single direction." Advances in Neural Information Processing Systems 37 (2024)
>
> [2] Yu, Lei, et al. "Robust LLM safeguarding via refusal feature adversarial training." ICLR (2024).
>
> [3] Zheng, Chujie, et al. "On prompt-driven safeguarding for large language models." ICML (2024).
>
> [4] Wollschläger, Tom, et al. "The geometry of refusal in large language models: Concept cones and representational independence." ICML  (2025).
>
> [5] Panickssery, Arjun, Samuel Bowman, and Shi Feng. "Llm evaluators recognize and favor their own generations." Advances in Neural Information Processing Systems 37 (2024): 68772-68802.
>
> [6] Chen, Guiming Hardy, et al. "Humans or llms as the judge? a study on judgement biases." EMNLP (2024).
>
> [7] Zhou, Zhenhong, et al. "On the role of attention heads in large language model safety." ICLR (2024).
>
> [8] Wang, Mengru, et al. "Detoxifying large language models via knowledge editing." ACL (2024).

---

> > ### Comment · Reviewer_747T · 2025-08-01
> >
> > 1. Your response makes sense, and I think you could make these points in the body or appendix. If you've conducted experiments that demonstrate that substring matching is a more robust measure of refusal than a frontier model as a judge (GPT4-o), I think that'd be valuable information to include, to justify your decision. I also maintain that you should explicitly provide the list of refusal phrases that you use, and how you obtained them.
> >
> > 2. I could buy your justification, as a sort of summary statistic of the observations in Figure 2. I suggest you add such a justification to the paper or appendix.
> >
> > 4. I still don't understand why the clusters are so neat here (not just GCG - all the clusters are very neatly separated; and I don't understand how these classes of prompts can be so cleanly separated, with hardly any overlap; I'd expect a more continuous spectrum - why is it seemingly discontinuous?). Also, that data size (20 each?) is still very small. I would appreciate if the authors could attempt to explain such clean separation in these plots. If my questions with this plot are addressed, I'd be open to raising my score.

---

> > > ### Author Response · Authors · 2025-08-04
> > >
> > > Thank you for your clear and prompt feedback on our response. We will follow your advice to make corresponding justifications and updates in our revision.
> > >
> > > ## Response to how to interpret the clustering in Figure 3 (b)
> > > **Why can there be clusters?** The x-axis ($\Delta_\text{harmful}$) and y-axis ($\Delta_\text{refuse}$) respectively represent the difference in cosine similarity between the hidden states of an input instruction and the centroids of two clusters. These values may reflect how the LLM perceives the harmfulness and refusal level of the instruction. For example, instructions refused by LLMs tend to have positive $\Delta_\text{refuse}$.  Instructions that share similar latent features associated with harmfulness or refusal can have similar cosine similarities with the corresponding cluster centroids. As a result, they may exhibit similar $\Delta_\text{harmful}$ or $\Delta_\text{refuse}$ values, forming distinct clusters in the plot.
> > >
> > >
> > > **Why is the clustering for examples of jailbreak methods especially evident?**
> > > 1. This is likely due to the shared structural patterns inherent in each jailbreak method. For example, in GCG attacks, all instructions are appended with the same *sequence* of adversarial suffixes (as the suffix is optimized over a batch of harmful data in GCG to generate similar response tokens, typically “Sure, I will…”). This may lead to similar model interpretations. Template-based jailbreak methods also use consistent prompting templates across different instructions.  The shared structure among prompts of the same jailbreak method may cause LLMs to perceive these examples as having a similar level of harmfulness or refusal in LLMs. As a result, points in a given jailbreak class tend to cluster within a specific region of the figure. Meanwhile, different jailbreak methods can vary significantly (examples of each jailbreak method are shown in Table 8 in the Appendix). This variability leads to dispersed clusters at different ranges of $\Delta_\text{refuse}$ and $\Delta_\text{harmful}$ across different jailbreak methods.
> > >
> > > 2. Persuasion-based jailbreak examples are more widely distributed. Some of the points of persuasion attack completely overlap with template-based jailbreak examples. But those examples of persuasion attack can still have similar patterns with each other, e.g., if using the same persuasion taxonomy [1]. So, some examples may still be interpreted by LLMs similarly in terms of harmfulness and refusal.
> > >
> > > 3. We also note that the visual neatness of the figure may be partially influenced by the scaling of the x- and y-axes, which were adjusted to accommodate the full range of jailbreak methods. Because the clusters of these jailbreak methods are quite dispersed, the figure is plotted with a sufficiently wide axis range to display them all together. If we zoom in on the region corresponding to a single jailbreak method by setting the axes limits, e.g., to the minimum and maximum of that method’s cluster, the points within that cluster will look more spread out.
> > >
> > >
> > > **Data size for points of GCG in Figure 3 (b).** For gcg, we observe the points especially cluster around the zero (e.g., the mean $\Delta_{\text{refuse}}$ across 200 examples is -0.066, and the standard deviation is 0.005). Adding more points to the figure may influence the clear visualization as they will overlap, so we did not do this for the current version.
> > >
> > >
> > > **Plain instructions in Figure 3 (a) are fuzzy along the axes of harmfulness or refusal.**  We would like to point out, as shown in Figure 3(a), that the points appear more spread out, though not necessarily along both axes at the same time.  For example, refused harmful examples demonstrate similar $\Delta_\text{refuse}$, while much more diverse $\Delta_\text{harmful}$ (e.g., many points have around 0.3 $\Delta_\text{refuse}$, and their $\Delta_\text{harmful}$ range from 0.05 to 0.15).  This suggests that these examples may belong to different types of harmful content. Some of them may not be strongly aligned with the most prototypical harmfulness features, yet all can trigger similarly strong refusal signals from the model.
> > >
> > >
> > > We hope that our explanations help address your confusion. We will incorporate these explanations into our revised manuscript. Please don’t hesitate to let us know if further clarification is needed. We are more than happy to elaborate. We believe our experiments are generally comprehensive and rigorous, and we sincerely thank you again for your thoughtful review and constructive feedback. We hope this addresses your concerns and leads you to a more positive evaluation of our work.
> > >
> > >
> > >
> > > [1] Zeng, Yi, et al. "How johnny can persuade llms to jailbreak them: Rethinking persuasion to challenge ai safety by humanizing llms." Proceedings of the 62nd Annual Meeting of the Association for Computational Linguistics (Volume 1: Long Papers). 2024.

---

> ### Author Response · Authors · 2025-08-06
>
> Thank you again for your review and important feedback! We are glad our clarification has solved most of your concerns and hope our new response can clarify your remaining question. We would be grateful if you would consider a higher score if our responses successfully address your concerns.

---

> > ### Comment · Reviewer_747T · 2025-08-06
> >
> > Thanks for your detailed response. I've raised my score.

---

### Official Review · Reviewer_LVR6 · 2025-06-27

**Clarity:** 2
**Significance:** 2
**Originality:** 2
**Rating:** 4
**Confidence:** 4

**Summary:**

This article studies LLMs’ internal perception of harmfulness, revealing that harmfulness and rejection responses are primarily encoded at different token positions and that LLMs may internally know the correct harmfulness level of instructions but still mistakenly accept or reject them.

**Questions:**

see weakness

**Ethical Concerns:**

["NO or VERY MINOR ethics concerns only"]

**Final Justification:**

The author's reply basically made me understand the core contribution of this paper and addressed my concerns.

**Limitations:**

yes

**Paper Formatting Concerns:**

no concerns

**Quality:**

2

**Strengths And Weaknesses:**

## Summary Of Strengths:

1. The article explores the recognition of harmful instructions in large language models (LLMs) and distinguishes how harmful and rejection responses are encoded at different token positions.
2. Based on the understanding of harmful recognition in LLMs, the article proposes a model called Latent Guard, which can effectively identify and reject harmful instructions while reducing false rejections of harmless instructions.

## Summary Of Weaknesses:

1. Unclear description of cluster centers: In Figure 2, it is not clear whether the cluster centers of $t_{inst}$ and $t_{post-inst}$ are calculated independently or use a unified value. If the same cluster center is used to calculate the cosine similarity of these two positions, the parameter distribution may be significantly affected, resulting in an inaccurate reflection of the actual phenomenon.
2. Specific mechanism of the inversion method: How to implement the inversion method mentioned in Section 3.5, especially how to effectively convert harmless content into harmful content instead of just adjusting the parameters to make the probability of generating logit from "certainly" to "no", can it be further clarified. Explanation is not simply changing the output tendency, which helps to provide a deep understanding of the difference between the harmfulness direction and the rejection direction.
3. Positive and negative sample clustering characteristics: I hope to see more precise data to prove that the harmful and harmless sample clusters shown in Figure 2 do have obvious distinguishing characteristics. Although the current graphics and analysis intuitively demonstrate this concept, there is a lack of specific quantitative evidence to support the effectiveness and stability of this distinction.

---

> ### Author Rebuttal · Authors · 2025-07-31
>
> 1. > Unclear description of cluster centers: … If the same cluster center is used to calculate the cosine similarity of these two positions, ….,  resulting in an inaccurate reflection of the actual phenomenon.
>
> Thanks for your question. We want to clarify that the cluster centers for $t\_{inst}$ and $t\_{post-inst}$ are **distinct** and computed independently. As emphasized in our paper, we analyze hidden states extracted specifically at $t\_{inst}$ and $t\_{post-inst}$ token positions, respectively. As described on Lines 156-158, the cluster centers at each of these positions are computed based on the hidden states *at that specific token position*. We will ensure our narrative in the revised manuscript makes this clearer to prevent any misunderstanding.
>
>
>
> 2. > ... How to implement the inversion method mentioned in Section 3.5, especially how to effectively convert harmless content into harmful content instead of just adjusting the parameters to make the probability of generating logit from "certainly" to "no" ...
>
> Thanks for your comments.
>
>  We designed the inversion task to provide causal evidence that **the harmfulness direction and the refusal direction are distinct internal representations** within LLMs. By demonstrating that steering the model using these two directions leads to different behavioral outcomes, we directly **challenge the common assumption in prior works [1, 3, 4] that harmfulness and refusal are conflated and identically encoded at $t\_{post-inst}$**. This distinction is crucial, as it suggests that current safety alignment techniques may operate on a superficial level; an LLM's refusal or acceptance does not necessarily reflect its true internal assessment of content harmfulness.
>
> **What we do:** We need to clarify that we are **not adjusting model parameters** but rather **steering the activations of input tokens**. This is a common and established method in the literature [1, 6, 7] to causally explain and manipulate latent concepts in LLMs. Specifically, we add the steering direction to the activations of input tokens at a given layer $l$, following the description in Section 3.4. We have added more details regarding these implementation specifics in the appendix of our new script.
>
> **What our results show:** Our results indeed demonstrate that **steering with our harmfulness direction can make LLMs internally assess harmless input content as harmful**. For example, steering hidden states along the harmfulness direction makes models answer "Certainly" to a harmless instruction (e.g., "Tell me how to make a cake. Does this user prompt cause harm?"), where the model initially answered "No" without steering. Conversely, steering with the refusal direction **could not flip** this "No" response to "Certainly." This indicates that the two directions are distinct: the harmfulness direction represents the concept of harmfulness that is manipulated in our steering experiments, while the refusal direction represents the direct refusal signal.
>
>
>
> 3. > ... I hope to see more precise data to prove that the harmful and harmless sample clusters shown in Figure 2 do have obvious distinguishing characteristics. Although the current graphics and analysis intuitively demonstrate this concept, there is a lack of specific quantitative evidence to support the effectiveness and stability of this distinction.
>
> Thanks for your comments. For our clustering analysis, we want to stress that our primary goal is **not to prove the distinctiveness of harmful and harmless sample clusters**, which has been shown in prior works [1,3,4,5]. Instead, we investigate how clustering patterns are formed to understand what information is encoded. Specifically, we demonstrate the **different clustering patterns** at $t\_{inst}$ and $t\_{post-inst}$, which suggests that harmfulness tends to be encoded at $t\_{inst}$, while the explicit refusal signal emerges and is encoded at $t\_{post-inst}$. We conduct our clustering analysis following [1] that computes the mean of a group of data as a cluster centroid. We then analyze which centroid an input is closer to. Given the inherent suggestiveness of clustering analysis, we provide **causal evidence** that harmfulness and refusal are encoded separately through our intervention experiments in the inversion task (Section 3.5).
>
> Following your comment, we computed the Silhouette Score to further support that the clustering is more influenced by whether the instruction is harmful rather than by refusal at $t_{\text{inst}}$. The Silhouette score ($S_\text{d} (A,B)$), ranging from -1 to +1,  quantifies how well an individual data point $d$ fits into a cluster $A$ relative to another cluster $B$. Positive numbers indicate the point fits well into this cluster, while negative numbers suggest the point might be assigned to the wrong cluster. Here are the layer-wise Silhouette Scores for "Refused harmless prompts" with respect to the "accepted harmless" cluster and "refused harmful" cluster at $t\_{\\text{inst}}$, and then with respect to the "refused harmful" cluster and "accepted harmless" cluster at $t\_{\\text{post-inst}}$:
>
> **$S_\text{Refused Harmless Prompts}$ (Accepted Harmless Cluster, Refused Harmful Cluster) at $t\_{\\text{inst}}$**
>
> ```
> [ 0.369,   0.5435,  0.4517,  0.3896,  0.364,   0.2517,  0.2273,  0.1915,  0.1694,
>   0.1797,  0.1736,  0.1621,  0.1678,  0.176,   0.1426,  0.1366,  0.1414,  0.1298,
>   0.1305,  0.1365,  0.1405,  0.1335,  0.1385,  0.1251,  0.1101,  0.1054,  0.1053,
>   0.1105,  0.1172,  0.1101,  0.1153,  0.1271 ]
> ```
>
> Interpretation: These positive scores indicate that at $t\_{\\text{inst}}$, "refused harmless prompts" are generally closer to the "accepted harmless" cluster than the "refused harmful" cluster. The first few layers have especially large scores, meaning the clustering is more evident. This suggests that harmfulness/harmlessness rather than refusal/acceptance decides the clustering at this stage.
>
>
> **$S_\text{Refused Harmless Prompts}$ (Refused Harmful Cluster, Accepted Harmless Cluster) at $t\_{\\text{post-inst}}$**
>
> ```
> [ 0.3408,  0.2063,  0.1457, 0.1576,  0.1808, 0.16934, 0.11604,
>   0.13403, 0.1953,  0.1466, 0.1334,  0.1942,  0.2502,  0.3455,  0.3125,
>   0.26,    0.2554,  0.2622,  0.2925,  0.2903,  0.3032,  0.29,    0.3245,
>   0.3076,  0.2952,  0.2793,  0.26,    0.2288,  0.2128,  0.1954,  0.1544,
>   0.1843 ]
> ```
>
> Interpretation: The *pattern* of clustering shifts to reflect refusal: these scores indicate that the prompts are now coherently grouped with "refused" examples, demonstrating the emergence and consolidation of the explicit refusal signal at this later stage.
>
> While higher scores indicate stronger clustering, it is important to note that scores in high-dimensional spaces like LLMs' hidden states are generally expected to be lower due to the *curse of dimensionality* and the inherent complexity of their representations [2]. However, the Silhouette scores still provide robust quantitative support for our central hypothesis: that **harmfulness tends to be encoded at $t\_{\\text{inst}}$, while the explicit refusal signal is encoded at $t\_{\\text{post-inst}}$**. Our work clarifies the assumption in past work that LLMs conflate harmfulness with refusal internally (harmfulness = refusal) at $t\_{\\text{post-inst}}$.
>
> If there are other specific types of quantitative data that would further clarify your concerns, we would greatly appreciate more concrete advice on what additional analysis or metrics would be most helpful.
>
>
>
> ## References
>
> [1] Arditi, Andy, et al. "Refusal in language models is mediated by a single direction." *Advances in Neural Information Processing Systems 37* (2024).
>
> [2] Trenton Bricken, Adly Templeton, Joshua Batson, Brian Chen, Adam Jermyn, Tom Conerly, Nick Turner, Cem Anil, Carson Denison, Amanda Askell, Robert Lasenby, Yifan Wu, Shauna Kravec, Nicholas Schiefer, Tim Maxwell, Nicholas Joseph, Zac Hatfield-Dodds, Alex Tamkin, Karina Nguyen, Brayden McLean, Josiah E Burke, Tristan Hume, Shan Carter, Tom Henighan, and Christopher Olah. "Towards monosemanticity: Decomposing language models with dictionary learning." *Transformer Circuits Thread*, 2023. [https://transformer-circuits.pub/2023/monosemantic-features/index.html](https://transformer-circuits.pub/2023/monosemantic-features/index.html)
>
> [3] Yu, Lei, et al. "Robust LLM safeguarding via refusal feature adversarial training." *ICLR* (2024).
>
> [4] Zheng, Chujie, et al. "On prompt-driven safeguarding for large language models." *ICML* (2024).
>
> [5] Jain, Samyak, et al. "What makes and breaks safety fine-tuning? a mechanistic study." *Advances in Neural Information Processing Systems 37* (2024): 93406-93478.
>
> [6] Li, Kenneth, et al. "Inference-time intervention: Eliciting truthful answers from a language model." Advances in Neural Information Processing Systems 36 (2023): 41451-41530.
>
> [7] Zou, Andy, et al. "Representation engineering: A top-down approach to ai transparency." arXiv preprint arXiv:2310.01405 (2023).

---

> > ### Comment · Reviewer_LVR6 · 2025-08-02
> >
> > Thanks, all my concerns have been addressed, and I think this paper is interesting.
> >
> > I will raise my rating. Good luck!

---

> > > ### Author Response · Authors · 2025-08-06
> > >
> > > Thank you very much for your review and feedback. We’re glad our responses have addressed your concerns. If possible, we would greatly appreciate it if you could consider adjusting the other scores accordingly.

---

### Official Review · Reviewer_P9bb · 2025-07-03

**Clarity:** 3
**Significance:** 3
**Originality:** 3
**Rating:** 5
**Confidence:** 3

**Summary:**

This paper identifies separate harmfulness and refusal directions by analyzing the hidden states at 2 different tokens in the prompt. They perform several behavioral experiments to verify the differing behavior between the directions they found.

**Questions:**

On line 216, between which vectors is the cosine similarity calculated? Is it the harmfulness direction in layer 9 and the refusal direction in layer 11, or the directions in layer 9, or the directions in layer 11?

When applying the steering directions, is it applied to all tokens from the beginning up until $t_{\text{post-inst}}$ for both the harmfulness and refusal directions, and on none of the output tokens?

Are the harmfulness and refusal vectors used in Sec 3.5 the same as the ones used in Sec 3.4? And are they applied to all of the additional tokens from the inversion prompting templates?

How does Section 5 compare with Section 6.2.1 of [Zou et al., 2023a]?

Andy Zou, Long Phan, Sarah Chen, James Campbell, Phillip Guo, Richard Ren, Alexander Pan,
Xuwang Yin, Mantas Mazeika, Ann-Kathrin Dombrowski, et al. Representation engineering: A
top-down approach to ai transparency. arXiv preprint arXiv:2310.01405, 2023a.

**Ethical Concerns:**

["NO or VERY MINOR ethics concerns only"]

**Final Justification:**

The authors clarified my potential concerns about their methodology. My evaluation of the novelty of the main takeaway remains unchanged. As a result I maintained my rating.

**Limitations:**

yes

**Quality:**

3

**Strengths And Weaknesses:**

This work applies existing methodology in interpretability to analyzing the harmfulness of instructions and whether a model will refuse it. Novel experiments separate out these two behaviors. This has significance in aiding existing efforts on model alignment.

---

> ### Author Rebuttal · Authors · 2025-07-30
>
> 1. > On line 216, between which vectors is the cosine similarity calculated? Is it the harmfulness direction in layer 9 and the refusal direction in layer 11, or the directions in layer 9, or the directions in layer 11?
>
> We compute the layer-wise cosine similarity between the harmfulness direction and the refusal direction. We then take the average across the layers to reduce the potential layer-wise bias.
> The result indicates two directions are geometrically different across all layers on average.
>
>
> 2. > When applying the steering directions, is it applied to all tokens from the beginning up until Tpost-inst  for both the harmfulness and refusal directions, and on none of the output tokens?
>
> Yes. Our steering implementation is different from past work [1] that intervenes with the input tokens and output tokens along the generation. We only intervene with the input tokens in Section 3.4. Our approach is designed to modify how the model internally views the input, allowing us to observe the resulting emergent output behaviors without direct manipulation of the generation process itself.
>
> [1] Arditi, Andy, et al. "Refusal in language models is mediated by a single direction." Advances in Neural Information Processing Systems 37 (2024)
>
> 3. > Are the harmfulness and refusal vectors used in Sec 3.5 the same as the ones used in Sec 3.4? And are they applied to all of the additional tokens from the inversion prompting templates?
>
> Yes. Those vectors are the same. We apply the harmfulness direction to all the tokens before the inversion question in the inversion prompting template so that we only change how the model views the instruction. But we find this implementation has minimum steering effects for refusal directions (e.g., refusal cannot be elicited by steering along the refusal direction), so for refusal direction, we still apply it to all the tokens in the whole input prompt.
> We have added more detailed explanation of implementations and supporting experiments in appendix.
>
>
> 4. > How does Section 5 compare with Section 6.2.1 of [Zou et al., 2023a]?
>
> Our latent guard in Section 5 is conceptually similar to  [Zou et al., 2023a]. They first compute a direction vector from $t_\text{post-inst}$ position and then use the dot product of hidden states with that direction vector to do binary classification.  However, we leverage harmfulness cluster and harmlessness cluster encoded at $t_\text{inst}$ for classification in our work and evaluate on more categories of input (over-refusal, jailbreak, etc) across different models more holistically. Because we have rigorously shown $t_\text{post-inst}$ mainly encodes refusal signals rather than harmfulness in LLMs, which is unreliable for classification. Although the limited evaluation of [Zou et al., 2023a] shows they can classify adversarial attacks with their classifier based on some threshold, the success may be from different strength levels of refusal signal in LLMs (weaker refusal signal than that of accepted harmless prompts). But their method may not apply to more test cases. For example, their method based on refusal direction is likely to fail for cases like persuasion attack, where the refusal signal is so weak that it is almost at the same level of accepted harmless instructions, as shown in our Figure 3 (a) and (b).

---

> > ### Comment · Reviewer_P9bb · 2025-08-07
> >
> > Thanks for the clarifications! I just have a couple small questions remaining.
> >
> > > We compute the layer-wise cosine similarity between the harmfulness direction and the refusal direction. We then take the average across the layers to reduce the potential layer-wise bias.
> >
> > Which layers did you take the average over? Is it all of the layers in the model?
> >
> > What were the layer-wise cosine similarities at layer 9 and layer 11, where the steering effects were strongest?

---

> > > ### Author Response · Authors · 2025-08-07
> > >
> > > Thank you for your review and feedback!
> > >
> > > We take the average over all layers. The cosine-similarity is 0.047 at layer 9, and 0.136 at layer 11.
> > >
> > > Thank you again.

---

> > > > ### Comment · Reviewer_P9bb · 2025-08-08
> > > >
> > > > I see, it appears the cosine-similarity is roughly the same even at the layers where the steering effects are strongest.
> > > >
> > > > Thanks for the clarifications! I'll maintain my rating. I recommend incorporating the clarifications into the paper itself.

---

### Author Response · Authors · 2025-08-01
**Significance and Novelty**

We sincerely thank all the reviewers for their valuable time and constructive feedback. We would like to first highlight why our conclusion that harmfulness and refusal are encoded separately is important. First, it suggests that the assumption of common works [1,2,3,4,5] is not correct that harmfulness and refusal are the same in LLM encoded at $t_\text{post-inst}$ (the last token of the post-instruction tokens). Second, our framework reveals that the current safety alignment technique for LLMs may be superficial that LLMs refuse instructions or accept instructions does not necessarily mean the model considers them as harmful or harmless. Furthermore, our framework reveals that encoding patterns can differ significantly across various token positions. This novel insight extends beyond analyses that default to using only the last token in prior works, opening new avenues for understanding LLMs' fine-grained internal beliefs and their influence on external behaviors.

### References

[1] Arditi, Andy, et al. "Refusal in language models is mediated by a single direction." Advances in Neural Information Processing Systems 37 (2024)

[2] Yu, Lei, et al. "Robust LLM safeguarding via refusal feature adversarial training." ICLR (2024).

[3] Zheng, Chujie, et al. "On prompt-driven safeguarding for large language models." ICML (2024).

[4] Sarah Ball, Frauke Kreuter, and Nina Panickssery. Understanding jailbreak success: A study of latent space dynamics in large language models. arXiv preprint arXiv:2406.09289, 2024.

[5] Zhihao Xu, Ruixuan Huang, Changyu Chen, and Xiting Wang. Uncovering safety risks of large language models through concept activation vector. Advances in Neural Information Processing Systems, 37:116743–116782, 2024.

---

### Decision · Program_Chairs · 2025-09-17

**Decision:**

Accept (poster)

**Comment:**

This paper shows that harmfulness and refusal are encoded separately in LLMs and introduces Latent Guard, which leverages harmfulness representations for more robust safety detection. Reviewers found the work novel and well-motivated, with convincing experiments and practical utility against jailbreaks. Main concerns were around evaluation metrics (substring-based refusal detection, cosine similarity choices), clarity in some sections, and generalizability to newer models, but these concerns were mostly addressed in rebuttal with additional analysis and clarifications. The consensus of final scores leans positive. I recommend acceptance.